# Cytokine Release Syndrome in the Immunotherapy of Hematological Malignancies: The Biology behind and Possible Clinical Consequences

**DOI:** 10.3390/jcm10215190

**Published:** 2021-11-06

**Authors:** Tor Henrik Anderson Tvedt, Anh Khoi Vo, Øystein Bruserud, Håkon Reikvam

**Affiliations:** 1Department of Hematology, Oslo University Hospital, 0372 Oslo, Norway; Totvedt@ous-hf.no; 2Department of Clinical Science, University of Bergen, 5020 Bergen, Norway; anh.khoi.vo@helse-bergen.no (A.K.V.); oystein.bruserud@helse-bergen.no (Ø.B.); 3Clinic for Medicine, Haukeland University Hospital, 5020 Bergen, Norway

**Keywords:** cytokines, chimeric antigen receptor, therapeutic antibodies, haploidentical allogeneic stem cell transplantation

## Abstract

Cytokine release syndrome (CRS) is an acute systemic inflammatory syndrome characterized by fever and multiple organ dysfunction associated with (i) chimeric antigen receptor (CAR)-T cell therapy, (ii) therapeutic antibodies, and (iii) haploidentical allogeneic stem cell transplantation (haplo-allo-HSCT). Severe CRS can be life-threatening in some cases and requires prompt management of those toxicities and is still a great challenge for physicians. The pathophysiology of CRS is still not fully understood, which also applies to the identifications of predictive biomarkers that can forecast these features in advance. However, a broad range of cytokines are involved in the dynamics of CRS. Treatment approaches include both broad spectrum of immunosuppressant, such as corticosteroids, as well as more specific inhibition of cytokine release. In the present manuscript we will try to review an update regarding pathophysiology, etiology, diagnostics, and therapeutic options for this serious complication.

## 1. Introduction

Cytokine release syndrome (CRS) was first described in the late 1980s as a systemic inflammatory response following treatment with anti-CD3 monoclonal antibody for graft rejection after solid organ transplants [1]. Initially used interchangeably with cytokine storm, a much broader term describing hyperinflammation caused by a large variety of disorders [2], CRS now refers to the immunological phenomenon triggered by immunotherapy such as, chimeric antigen receptor (CAR)-T cells [3], bi-specific T cell engagers (BiTEs) [4], or haploidentical allogeneic hematopoietic stem cell transplantation (haplo-allo-HSCT). However, as discussed in detail in a recent review there are many similarities between CRS secondary to these immunotherapeutic interventions and the COVID-19-associated cytokine storm syndrome; these authors therefore suggested that one should use the knowledge from the hematological CRS as a guideline for further studies, including clinical trials, in other cytokine storm syndrome, especially patients with critical severe acute respiratory syndrome coronavirus-2 (SARS-CoV-2) infections/coronavirus disease of 2019 (COVID-19) [5]. In the current article, we discuss etiology, pathophysiology, clinical manifestation, diagnostic approaches, and treatment modalities regarding CRS.

## 2. CRS Etiology

### 2.1. CRS Following CAR-T Cell Therapy

Treatment with tumor-antigen specific T cells genetically engineered to express CARs is highly effective in different cancers and has been approved in treatment of relapsed/refractory B-cell malignancies, including acute lymphoblastic leukemia (ALL) and multiple myeloma (MM). However, this approach can also result in severe toxicities that are directly linked to the induction of the potent immune effector responses. The massive cytokine release is believed to be caused both by the CAR-T cells themselves, bystander immune cells such as macrophages, as well as the tumor cells and their neighboring stromal cells. Several factors influence the severity of CRS. Firstly, a high number of infused T cells and the quality of the T cells significantly increase the risk of severe CRS [3,6]. Secondly, a high tumor burden may result in a significant degree of T cell activation and CRS [3,6]. Thirdly, to ensure adequate proliferation and maintenance of CAR-T cell must the CAR construct contain a contain co-stimulatory intracellular signaling domains CAR- T cells construct. Constructs utilizing, CD28 co-stimulatory domains have a higher risk of CRS than 4-1BB domain constructs; probably due to lower peak levels of T cell proliferation [7]. The risk of CRS also varies between different disorders and patients’ characteristics; with a significant higher incidence of CRS observed in patients with ALL compared with MM, and higher incidence of severe CRS in elderly patients compared with younger patients.

### 2.2. CRS Following Antibody Treatment

Monoclonal antibodies that directly target T cells carry the highest risk of severe CRS. This is illustrated by the phase I clinical trial of the CD28-targeting TGN1412 monoclonal antibody [8], where six healthy volunteers experienced severe CRS requiring intensive care treatment within few hours for participants [8]. Other monoclonal antibodies associated with high risk of CRS are the CD3-targeting moruomab and basiliximab, in addition to daclizumab that target the IL-2 receptor (CD25) [9,10,11,12].

A specific form of antibody therapy is treatment with bispecific monoclonal antibodies. Bispecific monoclonal antibodies are proteins containing two antigen binding domains, one that targes tumor cells and a second antigen binding domain that target T cells. This ensures tumor recognition by T-cell, T-cell activation, proliferation, and T cell mediated cytotoxicity. Currently have only a few BiTEs have been approved, e.g., blinatumumab for ALL, although several are in development, e.g., mosunetuzumab for follicular lymphoma (FL). While treatment with blinatumumab is associated with a high risk of CRS, this does not seem to be true for BiTEs for other hematological malignancies.

Alemtuzumab is a monoclonal antibiotic that targets CD52 expressed by lymphocytes, monocytes, and dendritic cells [13]. Intravenous administration is associated with a high frequency of immediate toxicity mainly related to first-dose reactions, including fever, rigor, and skin rash. However, subcutaneous administration is generally well tolerated with significantly lower risk of CRS. The first dose of alemtuzumab in transplant conditioning is usually administered as a full intravenous dose of 20 mg together with steroid prophylaxis, in contrast to the dose-escalation schemes often followed in chronic lymphocytic leukemia (CLL) treatment. Even though the subcutaneous route is associated with a better safety profile it is rarely used in graft versus host (GVHD) prophylaxis.

### 2.3. CRS and Anti-Thymocyte Globulin

Anti-thymocyte globulin (ATG) is currently wildly used as GVHD prophylaxis [14], and CRS has been reported for patients receiving ATG infusion [15]. ATG is a polyclonal antibody preparation derived either from horse or rabbit immunized with human thymocytes or the human Jurkat T cell leukemia cell line. Mild infusion-related reactions are frequent, but severe CRS, however, still self-limiting, can also be observed. Infusion-related reactions can be associated with a large variety of acute and delayed immunological reactions. Although the various ATG products currently used differ regarding antigen specificity, all products target antigens expressed on T cells, B-cells, macrophages, and antigen presenting cells as well as proinflammatory cytokines.

### 2.4. CRS Following Haploidentical Allogeneic Stem Cell Transplant

Haplo-allo-HSCT is a form of allo-HSCT where the donor only shares one human leukocyte antigen (HLA) haplotype with the recipient and is mismatched for a variable number of HLA genes. This significant HLA mismatch results in an early and excessive activation of alloreactive T cells that would result in severe and fatal GVHD without adequate measures [16,17,18,19]. The most frequently used protocol to ameliorate the HLA disparity is administration of two doses of cyclophosphamide between Day 3 and 5 post-transplant [16]. Posttransplant cyclophosphamide results in a targeted destruction of allo-reactive T cells, while spearing the graft versus leukemia reactivity and T cells responsible for viral immunity.

There is a significant risk of CRS for haploidentical transplantation utilizing T cell repleted granulocyte colony- stimulating factor (G-CSF) mobilized peripheral blood stem cell grafts [20]. CRS usually occurs during the first posttransplant days prior to administration of cyclophosphamide and typically manifests with fever. This is possibly caused by rapidly proliferating alloreactive T cells, and it resolves upon administration of cyclophosphamide. Most cases are mild and administration of acetaminophen, in addition to cyclophosphamide, is usually sufficient to control the symptoms. Abboud et al. reported that the overall incidence of severe CRS (i.e., Grade 3 or 4) was less than 15% and with some patients experiencing mild neurological symptoms or reduction in left ventricular functions [21]. A significant increase in interleukin (IL)-6 levels experiencing CRS, and all patients experiencing severe CRS responded within 48 h after administration of tocilizumab. Although CRS resolved quickly, long term survival due to treatment-related mortality was significantly higher in patients that experienced CRS.

Salas et al. reported that addition of a total dose of 4.5 mg/kg ATG prior to stem cell infusion seemed to ameliorate the risk of severe CRS, but this strategy was associated with graft failure for 15.6% of patients [22]. Ongoing studies will evaluate the effect a single dose of ATG in combination with posttransplant cyclophosphamide on the risk of CRS and graft failure [23].

## 3. Pathophysiology and Biomarkers

### 3.1. The Development of CRS Involves Various Cells and a Wide Range of Both Immunoregulatory and Angioregulatory Cytokines

Our current understanding of the CRS pathophysiology is incomplete and mainly based on patient serum cytokine profiles, autopsy studies, and animal models. Although the underlying mechanisms of CRS are complex and possibly differ between causes/patients, the final biological effects and clinical manifestations are similar. The most important organ involvements are reduced renal function, pulmonary edema, cardiac dysfunction with reduced cardiac output, activation of platelets and the coagulation factor cascade with secondary disseminated intravascular coagulopathy (DIC), and central nervous dysfunction with seizures and altered mental state [1,2,3,4,6,24].

A hallmark of CRS is endothelial dysfunction [6], activation of immunocompetent cells, including macrophages and natural killer (NK)-cells [24]. The main pathophysiological mechanisms in CRS due to CAR-T cell therapy are summarized in Figure 1 [1,2,3,4,6,24,25,26,27]. The initial event is supraphysiological activation of endogenous and/or infused T cells due to activating interactions with antigen-presenting cells, tumor cells or direct stimulation of T cells by the antigen-binding fragment (FAB) segment of an antibody. Such interactions result in a massive release of interferon-γ (IFN-γ) by the activated T cells. This is followed by the release of a large variety of other cytokines, the most important are IL-6, tumor necrosis factor-α (TNF-α), IL-10, granulocyte-macrophage colony-stimulating factor (GM-CSF) and CCL2. These mediators are released by bystander normal immunocompetent cells, such as macrophages, NK-cells, and endothelial cells. However, the mediator can also be released by activated T cells; and has also been demonstrated for circulating T cells derived early after allo-HSCT. Hence a broad cytokine response may thus be initiated even during the initial alloreactive response after allo-HSCT [28,29,30].

Macrophages care central in the development of CRS [31], and upon activation, macrophages release a wide range of cytokines including both interleukins, chemokines, and immunoregulatory mediators (Figure 1). Many of these are potent modulators of endothelial cell functions and cell migration. CRS shares clinical and biochemical characteristics with several syndromes associated with macrophage dysfunction, such as. hemophagocytic lymphohistiocytosis (HLH) and macrophages activation syndrome (MAS). Several animal models have demonstrated the importance of monocytes/macrophage activation, and that blockade of cytokines from macrophage, i.e., Il-6 and IL-1β, ameliorates CRS [25,27,32]. Several of the clinical hallmarks of CRS are believed to be directly linked to macrophages activation: (i) hyperferritinemia due to release of apoferritin from macrophages, (ii) hypofibrinogenemia by release of plasminogen activator inhibitor, (iii) fever due to the release of IL-6, IL-β, and TNFα, and (iv) cytopenias due to the extensive release of interferon-γ [25,27,32].

Toll-like receptors (TLRs) and the nuclear factor kappa-light-chain-enhancer of activated B cells (NF-κB) pathway are potent activator of monocytes/macrophages [33]. In addition, previous studies have shown that TLRs activation is involved in the development of cytokine storm as a complication to various diseases [34,35,36]. To the best of our knowledge the possible importance of TLRs, as well as other pattern-recognizing receptors, in the pathogenesis of CRS has not been studied in detail. However, several of these receptors can bind agonistic intrinsic ligands and may thus be ligated by molecules derived from cells that are damaged by or during immunotherapy [37,38]. Finally, TLRs have NF-κB as a downstream target, and NF-κB inhibition (e.g., by direct inhibitors or proteasome inhibitors [39]), should therefore be considered as a possible therapeutic strategy in CRS.

Monocytes include various functionally different subsets [31]; the peripheral blood levels of these subsets differ between allotransplant recipients and normal individuals and these subsets also differ in their kinetics of posttransplant reconstitution [40]. However, it is not known whether or how the balance between various monocyte subsets influences the risk of developing CRS after various forms of immunotherapy, e.g., intensive conventional therapy, autologous (auto-) or allo-HSCT [41].

### 3.2. Systemic Signs of Inflammation in CRS; Characterization of the Systemic Cytokine Responses and the Use of Soluble Mediators for Pretreatment Risk Evaluation

Systemic levels of cytokines and other soluble mediators correlates with of severe forms of CRS. As mention previously, clinical factors are important for this evaluation; two previous studies identified the following independent pretreatment risk factors for posttreatment development of severe CRS: high bone marrow tumor burden, concurrent infectious disease, lymphodepletion using cyclophosphamide and fludarabine, higher CAR-T cell dose, thrombocytopenia before lymphodepletion, and preparation of CAR-T cells without selection of CD8^+^ central memory T cells [3,6].

The study by Teachey et al. [24] showed that CRS is associated with an extensive cytokine release, including interleukins, chemokines, growth factors and immunoregulatory as well as angioregulatory cytokines (Table 1). As would be expected, the effects of CRS on the systemic mediator levels are most extensive for patients with the most advanced disease. Many of these cytokines can induce an acute phase reaction; e.g., IL-1, IL-6 and TNF-α, but other members of the IL-6 cytokine family can also induce this reaction [42]. Increased levels of acute phase proteins are therefore observed in CRS (Table 2). Increased levels of both C-reactive protein (CRP) and ferritin are seen during CRS, and the increase in ferritin levels can be as high as the levels seen in HLH [24,43,44,45]. In contrast, fibrinogen is also an acute phase protein, but in the CRS the systemic fibrinogen, it is usually low due to extensive release of plasminogen activator inhibitor from macrophages.

The overall results presented in Table 1 shows that CRS is characterized by a broad systemic cytokine response with increased levels of a wide range of functionally diverse cytokines. This is true especially for patients with advanced CRS that is usually defined as Stage 4/5 disease (Table 6, and further discussion in Section 4.3), and this staging is also referred to in Table 6. It should be emphasized that even for several cytokines showing highly significant differences there may be a considerable overlap between these two patient subsets, and the cytokine showing the highest fold-increase can differ between patients [46]. One can also see from the table that some cytokines are listed in both columns. First, the IL-6 levels are usually substantially increased, but even for this cytokine there is an overlap between patients with advances and less severe CRS and exceptional patients with only a relatively small increase/difference in IL-6 levels have been described [5,24,48,51]. For these exceptional patients IL-6 targeting therapy may not be the optimal treatment. Second, IL-5 levels have been investigated only in a few studies, and one of these studies described an increase in IL5 levels for patients with advanced disease that reached borderline significance (*p* = 0.017) and showed a large overlap between patients with advanced and less severe CRS [24] Third, as indicated in Table 1 the levels of Ang-1, Ang-2 and vascular endothelial growth factor (VEGF) are altered in patients advanced CRS; these effects alterations are not caused by a difference in the frequency of neurotoxicity because the levels of these mediators did not differ between patients with and without neurotoxicity [47] Finally, the heterogeneity of CRS patients with regard to cytokine levels can at least partly be explained by differences in tumor burden, i.e., higher levels in patients with large burden) [48], although the median age and/or the frequency of adult patients (and probably/possibly also the pre-CAR T cell chemotherapy) in the various studies differs [6,24,46,47].

Endothelial cell damage and capillary leak are clinical hallmarks of CRS (Figure 1). The endothelial activation and stress index (EASIX) is defined as [(creatinine level × lactate dehydrogenase (LDH)) level/peripheral blood platelet count], and this marker of endothelial activation has been validated in the CAR-T cell therapy setting [53]. This study included patients who received treatment with CAR-T cells, and the authors investigated the association between EASIX and the immune effector cell-associated neurotoxicity syndrome (ICANS) in a group of 171 patients treated with axicabtagene ciloleucel (axi-cel) for large B-cell lymphoma. Patients were tested before lymphodepletion. CRS Grades 2 to 4 were diagnosed in 81 patients (47%) and ICANS Grades 2 to 4 in 84 patients (49%). Three risk groups for neurotoxicity could be identified when EASIX was combined with ferritin. EASIX was also combined with both CRP and ferritin and again three risk groups for Grades 2 to 4 toxicity could be identified. Thus, common laboratory parameters including endothelial markers and acute phase proteins correlate with CAR-T cells related toxicities.

A wide range of various cytokines, including chemokines, are important for regulation of vascular functions and the endothelial cell status [54]. The systemic levels of such mediators in CRS have been investigated only for some of them, and these observations further support the hypothesis that endothelial and vascular modulation is important in the pathogenesis of CRS [24]. Angiopoietin(Ang)-2 and von Willebrand factor (vWF) are both regarded as markers of endothelial activation, and systemic levels of both these markers are increased in patients with severe CRS both before lymphodepletion (i.e., signs of endothelial activation before the treatment) and during the development of CRS [6] (Figure 1).

Neurological involvement in CRS is associated with abnormalities in the cerebrospinal fluid including increased levels of white blood cells, protein, IFN-γ, IL-6, IL-10, and granzyme B [47]. Furthermore, the cerebrospinal fluid level of the glial fibrillary acid protein is regarded as a marker of astroglial injury whereas the s100 calcium binding protein level indicates astrocyte activation; the spinal fluid levels of both these markers are increased during in CRS with neurotoxicity.

### 3.3. The Central Role of IL-6 and Angiopoietins in CRS

Increased serum IL-6 levels are a hallmark of CRS, and IL-6 blockade effectively ameliorates most symptoms of CRS [24,55,56,57,58,59,60]. IL-6 exhibits a wide variety of biological effects through classical and IL-6 trans-signaling. Only a limited number of cell types, e.g., naive T cells, hepatocytes, monocytes and neutrophils, express the IL-6 binding IL-6 receptor (IL-6R) chain. The IL-6R is then expressed together with the signal-transducing gp130 transmembrane glycoprotein and these cells can thereby respond to IL-6 alone; this is referred to as classic IL-6 signaling. All other cell types do not express membrane-bound IL-6R and for this reason IL-6 alone cannot initiate intracellular signaling. However, most cells express the gp130 signal-transducing glycoprotein, and these cells can bind and thereby respond to the soluble complex of IL-6 and IL6R. This IL-6/IL-6R initiation of intracellular gp130-mediated intracellular signaling is referred to as IL-6 trans-signaling. This trans-signaling is thus regulated by the release of soluble IL-6R. Trans-signaling is believed to contribute to the development of many CRS symptoms and signs, i.e., disseminated intravascular coagulation, vascular leakage and myocardial dysfunction [57,58,59]. The peak systemic levels of IL-6, soluble IL-6 receptor, IFN-γ, and soluble gp130 directly correlate with the risk of developing severe CRS [24].

A recent study demonstrated that severe CRS coincided with elevated serum levels of vWF and Ang-2, both released from Weibel–Palade bodies on endothelial activation [6]. Furthermore, high serum vWF level and Ang-2: Ang-1 ratio were observed prior to CAR T cell therapy for patients who developed more severe CRS. Lastly, severe CRS was also associated with thrombocytopenia before lymphodepleting chemotherapy, possibly due to the lack of the endothelial stabilizing cytokine Ang-1, of which thrombocytes are a main source [6]. Taken together, these findings suggest that preexisting and excessive endothelial activation might be risk factors for severe CRS.

### 3.4. Potential Biomarkers in CRS

As described above CRS is associated with altered systemic levels of a wide range of soluble mediators (Table 1 and Table 2). Non-specific markers of inflammation such as CRP and ferritin are obligate elevated in CRS and correlate with the disease severity. However, they fail at predicting the occurrence of severe CRS [24,47]. Teachey et al. identified a series of cytokines significantly elevated in CRS that that correlated with the occurrence of severe CRS. By evaluating the concentrations of IFN-γ, IL-13, and CCL3 in pediatric patients, this model achieved a sensitivity of 100% and specificity of 96%. This was subsequently validated in an independent cohort of 12 pediatric patients [24]. However, their predicting models evaluate the cytokines within 72 h after infusion, which may diminish its clinical value as severe CRS frequently occurs earlier and often within 72 h after infusion [2,6,61]. A timelier method was reported by Hay et al.; evaluation of CCL2, measured in patients with temperature >38.9 °C and within 36 h of infusion, was found to be superior to CRP, ferritin, and other cytokines in predicting severity CRS [6]. Simple predictive models using the combination of clinical parameters and biomarkers may be more suited for routine clinical practice, as testing for the aforementioned cytokines are not commonly available. Further studies are needed to validate and develop viable predictive models.

As stated above the EASIX index has also been investigated as a possible biomarker in combination with CRP and ferritin [53]. This index could identify three different patient subsets regarding development of CRS Grades 2–4 with 74%, 51%, and 29% cumulative incidences. This index should also be suitable for routine clinical practice, although it can be questioned whether the combination with CRP/ferritin is helpful because the classification into three groups and the corresponding cumulative indices are very similar to the results when using the original index alone.

In our opinion, there are several possible additional mediators that should be further investigated as possible biomarker in CRS. Especially, specific metabolites previously been shown to correlate endothelial cell dysfunction, capillary leak and or altered renal functions, and factor of complement system due to the crucial role of various complement fragment to initial cell migration and vascular leakage.

First, several metabolites are associated with capillary leak and or altered renal functions and due to the importance of endothelial cell dysfunctions in CRS these markers should also be investigated in CRS [62]. Second, there is activation of the complement system [63,64]. Finally, more detailed investigations of immunocompetent cell subsets should be investigated, especially monocyte subset distribution and monocyte activation that is important in endothelial/vascular biology [65,66].

The possible use of soluble adhesion molecules as biomarker in CRS should also be further investigated [67,68,69,70]. The systemic levels of several adhesion molecules derived from both immunocompetent and endothelial cells can be released during inflammation, these molecules show biological activity, they can have immunoregulatory functions and may also be involved in regulation of coagulation. Their possible role as biomarkers in CRS should be further investigated, e.g., biomarkers for endothelial activation.

### 3.5. The Biological Heterogeneity of CRS Patients

Patients with CRS seem to be heterogeneous with regard to certain cytokines, although this heterogeneity seems to be limited. First, most studies describe increased IL-6 levels, but some exceptional patients/studies do not show increased systemic IL-6 levels [24,48]. Second, both increased and unaltered levels have been described for IL-5 [24,47], the same is true for detection of coagulopathy [6,48]. Finally, the predictive value of early CRP levels (i.e., first three days after CAR-T cell infusion) has not been seen in all studies either [24,47]. This possible heterogeneity between patients’ needs to be further investigated, especially emphasizing the possible relevance for diagnosis and the choice of cytokine-targeting treatment.

Development of an acute phase reaction seems to be a common characteristic in CRS, but this reaction can be initiated by various cytokines including IL-1, TNF-α, and IL-6 as well as other members of the IL-6 family [43]. Although certain cytokines seem to be more important than others and therapeutic targeting of single cytokines is often effective, in our opinion, it is most likely that the acute phase reactions reflect the contribution of several cytokines to CRS development. For this reason, one should also investigate the possible diagnostic value of extended cytokine profiles in addition to single cytokine levels in future clinical studies. This diagnostic approach would be similar to the approach used in recent studies of human cancer; CRP has a prognostic impact both in renal cancer and head and neck squamous cell carcinoma, but the acute phase cytokine profile differs between them [71,72]. Thus, the molecular mechanisms behind the acute phase reaction differ between these two cancers, and a similar heterogeneity may also exist for CRS patients.

### 3.6. The Lessons from Studies of Animal Models

Two animal models of CRS following CAR-T cell therapy have recently been published [25,27]; the main characteristics of these models and the most important observations are summarized in Table 3. First, the importance of host macrophages in the development of CRS was demonstrated in both studies. Second, therapeutic targeting of both IL-1 and IL-6 could be effective, but the two therapeutic strategies differed regarding their clinical effects. This targeting did not influence the antileukemic effects. Finally, other therapeutic targets were also suggested, including CD40-initiated signaling and targeting of nitric oxide synthase. Thus, both these models seem relevant for future studies of potential new biomarkers (e.g., soluble adhesion molecules, metabolites, extended cytokine profiles) and new therapeutic strategies/targets.

## 4. Clinical Evaluation of Patients with Suspected CRS: Symptoms and Signs, Diagnostic Work up, Differential Diagnoses, and Grading

### 4.1. Clinical Manifestations

The onset and peak of CRS generally occurring in the first week after therapy. Clinical manifestations of CRS vary (Table 4) from mild, flu-like symptoms to severe life-threatening multiorgan failure secondary to an uncontrolled inflammatory response. High fever is seen for most patients, although not all [73,74]. Mild symptoms of CRS include fatigue, rash, arthralgia, and myalgia, whereas more severe cases are characterized by vasodilatation with subsequent hypotension; this can develop into an uncontrolled systemic inflammatory response with vasopressor-demanding vasodilatory shock and severe vascular leakage [73,74]. Furthermore, a cytokine induced cardiac dysfunction is sometime observed as a rapidly progressing cardiomyopathy with clinical features of stress cardiomyopathy (i.e., Takotsubo syndrome) [75]. Cardiac dysfunction, neurologic toxicity, renal failure, hepatic failure and DIC, can all be seen in fulminant CRS [73,74]. CRS is usually reversible.

Central nervous system (CNS) symptoms are sometimes seen in conjunction with CRS. However, due to a possible different pathophysiological mechanism neurologic toxicity that is observed after immunotherapy, is termed either immune effector cell-associated neurotoxicity syndrome (ICANS) or cytokine release encephalopathy syndrome (CRES). Symptoms range from headache, memory difficulties, diminished attention, and language disturbance to confusion, delirium, aphasia, motor weakness, myoclonus, seizure and signs of cerebral edema [73]. Although most cases of neurological toxicity are reversible, life-threatening cerebral edema in patients with CRS has been reported. Other symptoms may also occur and are classified in the table below according to the affected organ system (Table 4).

### 4.2. Diagnostic and Differential Diagnoses

There are currently no single diagnostic tests that can differentiate CRS from sepsis, allergic reactions or HLH. Common laboratory abnormalities in patients with CRS include cytopenia, elevated creatinine and liver enzymes, disturbed coagulation parameters, and high CRP levels. As described above CRS is usually associated with high levels of interferon-γ IL-6, TNF-α and IL-10 [6]. However, currently there are no clear correlation between cytokine levels, disease severity and outcome. High ferritin levels indicate macrophage activation or a HLH similar to clinical picture. Currently, CRS, remains an exclusion diagnosis and several differential diagnoses have to be excluded (Table 5). Infections have to be excluded by adequate microbiological examination including blood and urine cultures together with sampling from relevant organs systems [76]. Evaluation of left ventricular function by echocardiography is required in patients with sign of cardiac dysfunction [75]. Radiological examinations preferably computed tomographic (CT) with intravenous contrast, and bronchial lavage are required to adequate rule out other cause of acute respiratory distress syndrome (ARDS). 

CRS may also be associated with signs consistent with MAS or HLH [77]. MAS/HLH is a hyperinflammatory syndrome that shares features and is probably related to CRS. Both CRS and HLH include signs of macrophage activation and cytokine storm [77]. Examination of ferritin, triglycerides, fibrinogen and soluble interleukin-2 receptor (soluble CD25) can provide diagnostic information regarding HLH [77]. However, most patients with moderate to severe CRS have laboratory results that meet the classical criteria for HLH/MAS, although hepatosplenomegaly, lymphadenopathy, and overt evidence of hemophagocytosis are less common. Both CRS and HLH show increased levels of IFN-γ together with T cell activation, but the main cause of T cell activation in HLH is believed to be insufficient regulation of T cells by absent/reduced NK-cell function. Furthermore, a well-defined genetic predisposition can trigger the cytokine storm in HLH, but the secondary forms of HLH are less well defined and especially for haplotransplant recipients a clear distinction between CRS and HLH is not always possible.

Tumor lysis syndrome (TLS) may occur coincidently together with CRS because of massive immune cell activation/expansion and a strong anticancer effect. The patients should therefore be evaluated for the typical disturbance associated with TLS; i.e., hyperuricemia, hyperpotassemia, hyperphosphatemia and hypocalcemia [64]. Furthermore, tumor/leukemia progression, in the setting of relapse or refractory disease, can also occur and should be kept in mind by uncharacteristic and progressive symptoms. Finally, allergic reactions, especially severe drug reactions, can cause symptoms resembling CRS with fever and hypotension, and work up regarding the patient’s recent medications should be performed.

### 4.3. Grading of CRS

Several grading systems for CRS have been developed; for this reason, the grading systems applied in various studies therefore vary widely, making comparisons between different studies/clinical trials difficult. Furthermore, in an initial definition CRS onset was defined as within 24 h after initiation of therapy, but later studies have shown that this is not always true and should not be used as a criterion, especially not for CRS associated with CAR-T cells and other cell therapies. In addition, the previous grading as included as a criterion, whether the drug infusion was interrupted, an approach that is less relevant for cellular therapy.

The Lee criteria [73] established in 2014 have been regarded as the most relevant criteria for clinical practice (Table 6). However, a new grading system has recently been proposed by the American Society for Transplantation and Cellular Therapy (ASTCT) with the aim of providing a uniform consensus grading system for CRS (Table 7) [74]. These new criteria are only based on fever, hypotension and hypoxia (Table 7) In this scoring system the CRS grade is determined by the most severe event in CRS, i.e., fever, hypotension or hypoxia that cannot be attributed to any other cause. For example, a patient with a temperature of 39.5 °C, hypotension requiring one vasopressor, and hypoxia requiring low-flow nasal cannula is classified as Grade 3 CRS. This scoring system relies on well-defined clinical data and is relatively easy to interpret. The aim of this scoring system is to allow comparison across different sites and clinical trials, and thereby to facilitate optimal strategies for prevention and management of CRS. Finally, because immune effector cell-associated CRS can be associated with high morbidity and mortality rates if not recognized and treated properly, the new CRS grading was developed to accurately captures these early features of the condition [74]. A staging system only based on clinical parameters can also be used for all patients independent of the possible patient heterogeneity with regard to the pathogenesis, proinflammatory signaling and differences in organ involvement. 

### 4.4. Heterogeneity of CRS Patients; the Clinical Evidence

CRS is a heterogeneous disease. First, as described in Section 3.5 there is biological evidence for CRS heterogeneity, e.g., descriptions of exceptional patients with regard to the systemic cytokine profiles, even patients with normal IL-6 levels. Second, CRS can be caused by very different immunotherapies (e.g., CAR-T cells, allo-HSCT). One would also expect that differences in previous anticancer chemotherapy (type, intensity and duration of chemotherapy) would influence the pretreatment immunological status of these patients and thereby modulate the pathogenesis of CRS [78]. Third, one would in addition expect age-dependent differences in the immune system to influence the risk and/or development of CRS [79]. Finally, development of CRS is associated with many different risk factors (including several clinical risk factors), and it would not be surprising if these very different biological and clinical factors mediated their effects through different molecular mechanisms.

## 5. Treatment

### 5.1. General Suggestions

The treatment of CRS in allo-HSCT recipients has been investigated in very few clinical studies. This is in contrast to CAR-T and BiTE studies, where the treatment is usually based on the grading of the syndrome (see Section 3.4) and on the general experience from patients with other causes of CRS. 

Patients with Grade 1 CRS (Table 6) are regarded to have symptoms that are not life threatening, and hence supportive and symptomatic therapy should be the main focus for this patient group [73]. This can include treatment for fever, nausea, and pain (e.g., headache, myalgia). Supportive care is thus needed, but close monitoring, including fluid balance and organ functions, and response evaluation is also important regarding reclassification of the patients based on the response to the initial fluid/vasopressor treatment. A continued evaluation with a focus on development of complicating infections is necessary both for these patients as well as patients with more advanced CRS stages. The endothelial involvement in CRS with extensive capillary leak may limit the response to fluid therapy alone [6]; and previous studies suggest that extensive posttransplant fluid retention/capillary leak in allotransplant recipients is associated with an adverse prognosis [62,80].

Grade 2 (Table 6) has been defined as hypotension that can be handled with fluid or only low-dose treatment with one vasopressor, mild respiratory symptoms responsive to low-flow oxygen, or Grade 2 organ involvement [73]. Patients with Grade 2 should be considered for immunosuppressive treatment, whereas patients with Grade 3 or more should receive immediate treatment [81]. The decision whether to start with immunosuppressive treatment should probably be based on a general clinical evaluation, and immunosuppression should be started early especially for elderly patients and patients with comorbidities who are judged not to be able to tolerate the altered hemodynamics and/or organ involvement associated with the syndrome. Independent of whether immunosuppressive treatment is initiated these patients should receive vigilant supportive care and close monitoring as described above for patients with Grade 1 disease. One should also remember that the potentially life-threatening complications in CRS are cardiac dysfunctions, respiratory distress syndrome, neurological toxicity, severe renal or hematological failure, and severe coagulopathy [73,74]. One should also keep in mind that close monitoring is necessary for Grade 2 patients regarding the dose of vasopressors needed for treatment of hypotension; the need for high-dose or multiple vasopressor treatment is a Grade 3 criteria as defined in Table 6 [73].

### 5.2. IL-6 Targeting Therapy

Tocilizumab is regarded as the first-line treatment by several authors [74,82]; this is a monoclonal antibody that binds to both soluble and membrane-bound IL-6 receptors and thereby prevents IL-6 binding to its receptor [82]. IL-6 inhibition is an effective treatment in many patients and response rates up to 70% in patients with severe CRS [82], the approve treatment for patients above 2 years of age is 8 mg/kg for adults and 12 mg/kg for patients with less than 30 kg body weight; the maximal single dose should be 800 mg and the interval between consecutive doses should be at least 6 h [82]. Many patients respond to this treatment within few hours or within two days; if no effect is seen within 24–72 h a second administration is feasible [82]. Other authors recommend a second dose and/or a second immunosuppressive agent already after 24 h if the patient’s condition has not improved or stabilized within 24 h [73]. Some aspects have to be emphasized with regard to tocilizumab. First, IL-6 is a driver of the CRP response and for this reason CRP is not a reliable biomarker for the severity of the cytokine release syndrome after administration of IL-6 targeting treatment [73]. Second, tocilizumab side effects may overlap with the clinical picture of CRS, e.g., transaminitis, thrombocytopenia (neutropenia seems less common), altered cholesterol/lipoprotein serum levels [73]. Finally, IL-6 levels can show a transient increase following tocilizumab administration [73].

Alternative strategies for IL-6 targeting could be siltuximab or clazakizumab; both these monoclonal antibodies bind to IL-6, thereby inhibiting classical and IL-6 trans-signaling [82].

### 5.3. Corticosteroids and Other Alternative Therapeutic Strategies

Corticosteroids are also effective in the treatment of CRS. Since corticosteroid inhibits T-cell proliferation and thereby possibly T-cell survival, it was believed that treatment with corticosteroids could hamper the T-cell mediated antitumor effect. Corticosteroids was therefore regarded as a second line therapy [73]. This recommendation/suggestion was however, based on a limited clinical experience with steroids and the fact that tocilizumab is often effective, responses are often rapid, and the general immunosuppressive effects are regarded as less severe with regard to both anticancer immune reactivity and risk of complicating infections [73]. In more recent CAR-T and BiTe studies, steroids are used to a greater extent and earlier in the CRS treatment algorithm with no apparent loss of anti-tumor effect.

It should also be emphasized that the clinical experience in allotransplant recipients is limited, and the clinical experience from patients with other causes of cytokine release syndrome may not necessarily be relevant for the priority between tocilizumab and corticosteroids as the first-line treatment in allotransplant recipients. It may also be relevant to consider combined treatment in patients with severe, life-threatening disease [73]. The dosing and choice of steroid should probably be individualized and adjusted to close monitoring of the patients, but commonly used doses have been methylprednisolone 2 mg/kg/day that is weaned over several days or dexamethasone 0.5 mg/kg with a maximal dose of 10 mg/dose [73,83]. It may be relevant to prefer dexamethasone in patients with neurological symptoms due to its more efficient penetrance to the central nervous system [42,47,74].

As described more in detail in a recent review the acute phase reaction is initiated and strengthened through the release of several cytokines and not only IL-6; other IL-6 family members as well as IL-1β and TNF-α can contribute to the development of the acute phase reaction and the cytokine release syndrome [82]. The cytokine release syndrome should be regarded as an extreme inflammatory response including an extreme acute phase reaction with the increase of several cytokines including increased systemic levels of IL-6, other IL-6 family members, IL-1β, TNF-α as well as various proinflammatory CCL and CXCL chemokines [82]. This assumption is further supported by the high CRP levels and the clinical picture that may fulfill the criteria for sepsis even in the absence of an infection [73,74,82]. Furthermore, clinical studies of CRS patients also suggest that additional cytokines are involved in the development of this complication [24]. Finally, the cytokine release syndrome is regarded as a non-antigen specific toxicity due to high-level immune activation [73]; it involves several immunocompetent or immunoregulatory cell subsets including T cells, monocytes/macrophages and endothelial cells (see Figure 1) and all these cells can release or respond to a wide range of cytokines when they become activated (e.g., IL-1α/β, TNF-α, IL-6, other IL-6 family members, CCL and CXCL chemokines) [28,29,33,84,85,86,87]. For these reasons, other molecular mechanisms in the development of the acute phase reaction/CRS may become relevant in the CRS treatment in allotransplant recipients, including TNF-α neutralizing antibodies (infliximab), soluble TNF-α receptors or IL-1 receptor-based inhibitors (recombinant IL1 receptor antagonist, anakinra) [73,74]. Such therapeutic strategies have clinically relevant effects in HLH and MAS [73,74,88]; and this experience may be relevant because CRS may fulfill the criteria even for these two inflammatory diseases at least in certain patients [73,74,82,89]. IL-6 may also promote the development of dysfunctional cytotoxic cells in CRS [90]; this represents an additional functional similarity between cytokine release syndrome and HLH/MAF.

## 6. Prognosis

The possible prognostic impact of CRS has been investigated only in a few studies. It may have a prognostic impact in allotransplant recipients through its association with later development of graft versus host disease, and the immunosuppressive treatment may also interfere with the antileukemic immune reactivity after allotransplantation.

Previous studies have demonstrated that pretransplant signs of inflammation with increased CRP levels are associated with adverse prognosis in allotransplant recipients [80], and it would not be surprising if development of a severe inflammatory response early posttransplant had a similar effect. This is further supported by a recent study suggesting that there is an adverse prognostic impact of pretransplant as well as early posttransplant IL-6 levels in patients receiving haploidentical allotransplantation [91]. Finally, patients with CRS (mainly low-grade CRS) after haploidentical transplantation seem to have an increased frequency of severe acute GVHD, although these authors could not detect any association between previous CRS and non-relapse mortality or overall survival [81,92]. 

A recent study described a very high frequency of posttransplant CRS in haploidentical stem cell transplantation, but severe CRS (i.e., Grades 3–5, see Table 6) occurred in 15–20% of their patients and these patients had a mortality of 24% [92]. Patients with severe CRS also had increased non-relapse mortality (38 vs. 8%). Severe CRS was most common in elderly patients and patients with a history of radiation therapy. Shorter median survival for haploidentical allotransplant recipients with severe CRS was also described in another recent study; these last authors also described later neutrophil reconstitution for CRS patients and tocilizumab therapy seemed to be well tolerated [21]. Both these studies were relatively small and the possible prognostic impact of IL-6 targeting therapy on the risk of later relapse could not be investigated. 

To conclude, severe CRS after allotransplantation is uncommon, has a high mortality and seems to be associated with increased overall non-relapse mortality. However, additional studies are needed to better clarify the prognostics impact, especially with regard to posttransplant antileukemic activity and relapse risk. 

## 7. Conclusions

CRS is a potentially life-threatening complication after various forms of T cell-based immunotherapy. Even though the clinical context is very different, these patients usually have in common a strong acute phase reaction and increased systemic levels of a wide range of cytokines. Several organs can be affected, and the severity differs between patients. In our opinion future clinical studies have to include more detailed studies of the molecular and cellular mechanisms behind the development of the syndrome. A generally accepted standardization of the patient staging/classification regarding severity is then required. There should in addition be a focus on patient heterogeneity not only with regard to risk of CRS but also with regard to molecular mechanisms and whether the treatment for individual patients should be based on targeting of selected cytokines (e.g., IL-1 and/or IL-6) or a general anti-inflammatory treatment (e.g., steroids). It will also be important to study possible late effects of this syndrome and the quality of life for patients who have developed this complication after anti-cancer therapy [93].

## Figures and Tables

**Figure 1 jcm-10-05190-f001:**
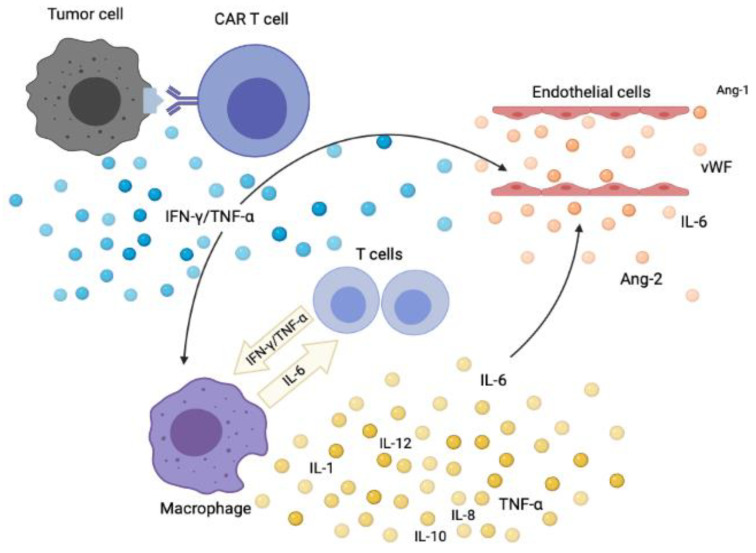
The development of CRS after CAR-T cell therapy. The triggering mechanisms are incompletely understood, but CRS is triggered by the release of immunostimulatory cytokines (e.g., IFN-γ, TNF-α) both from the tumor cells and CAR-T cells. Immunocompetent cells as well as stromal cells (e.g., endothelial cells) are subsequently activated and thereby release excessive amounts of proinflammatory cytokines as the final part of a positive feedback loop.

**Table 1 jcm-10-05190-t001:** The systemic cytokine profile cytokine in patients with cytokine release syndrome; a summary of serum/plasma levels for soluble mediators, soluble receptors and soluble adhesion molecules [6,24,46,47,48,49,50]. If a mediator is listed in both groups, this means that the results are conflicting (see comment in the text). Levels are generally increased; decreased levels are stated in the table. Most studies define advanced disease as Stage 4/5 (see Table 6 and further discussion in Section 4.3).

Increased in Advanced CRS	No Difference between Low and Advanced Stage CRS
**Interleukins**	
IL1-RA, IL-4, IL-5, IL-6, IL-8, IL-10 sgp130, sIL-1R1, sIL-1R2, IL2-RA, sIL-6R	IL-1β, IL-2, IL-5, IL-6, IL-7, IL-12, IL-13, IL-15, IL-17, IL-4R
**Chemokines**	
CCL2, CCL3, CCL4 CXCL9, CXCL10, CX3CL1	CCL5, CCL24
**Immunoregulatory mediators**	
IFN-α, IFN-γ, TNF-α	
**Growth factors**	
G-CSF, GM-CSF, Flt3-ligand	EGF, HGF, EGFR, VEGFR1, VEGFR2, VEGFR3
**Angioregulatory mediators**	
VEGF, Ang-2, von Willebrand factor Decreased Ang-1 levels.	Observed differences are not caused by increased frequency of neurotoxicity because no association with neurotoxicity and serum levels of VEGF-A, Ang-1, Ang-2 could be detected.
**Others**	
RAGE, Granzyme B	sCD90 No association with neurotoxicity and serum levels of International normalized ratio, D-dimer, vWF, nadir fibrinogen levels.

Abbreviations: Ang, angiopoietin; EGF, epithelial growth factor; G-CSF, granulocyte colony-stimulating factor; GM-CSF, granulocyte-macrophage colony-stimulating factor; HGF, hepatocyte growth factor; IFN, interferon; IL, interleukin; R, receptor; s, soluble; RAGE, receptor for advanced glycation end products; VEGF, vascular endothelial growth factor; vWF, von Willebrand factor.

**Table 2 jcm-10-05190-t002:** Systemic signs of inflammation in patients with cytokine release syndrome; systemic levels of selected acute phase proteins and cytokines.

Mediator	Systemic Level in CRS	Refs
**CRP**	High levels correlate with CRS severity (i.e., staging, see below). A rise is seen in most patients after CAR-T infusion compared with baseline levels. High initial levels cannot be used to predict the development of severe CRS. The initial increase/level cannot predict the clinical course (i.e., the stay at intensive care unit) in patients with severe disease. One study showed that high early levels could predict later severe disease; this has not observed in other studies.	[24,43,44,46,52]
**Ferritin**	A rise is seen in most patients after CAR-T infusion compared with baseline levels. Increased levels in CRS, the levels correlate with severity/staging and levels exceeding 10,000 mg/100 mL can be seen both in adults and children even in Stages 0–3 (for staging, see below). The level at transfer to intensive care unit has only a weak association with the length of the stay at the unit.	[24,43,44]
**Fibrinogen**	Although an acute phase protein, fibrinogen levels are either low (especially in children with severe disease) or normal due to release of plasminogen activator inhibitor from macrophages.	[24]

**Table 3 jcm-10-05190-t003:** Animal models for CRS; description of two various models and the most important observations [25,27].

Giavridis et al. [25]	Norelli et al. [27]
**Design of the model**	
Immunocompromised mice Intraperitoneal injection of Burkitt lymphoma Rajiv cells and NALM-6 pre-B ALL cells; mice were tested when vascularized tumors had developed intraperitoneally. CD18 recognizing CAR-T cells (human 1928 CAR-T cells)	Immunocompromised mice Human cord blood hematopoietic stem and progenitor cells were injected into the liver to reconstitute human hematopoiesis and development of immunocompetent cells. T cells from the mice were reconstituted with anti-CD44w6 or CD1920z CARs. THP1 and BV173 cells were used, in addition they used an ALL-CML cell line derived from a patient with CML in lymphoid blast phase and transfected with CD44 isoforms. These malignant cells were infused intravenously.
**Local cellular responses**	
CAR-T cell recognition of the malignant cells Tumor infiltration of myeloid cells	CAR-T cells had antileukemic effects; recognized the specific CD44 isoform and had a durable antileukemic effect.
**Systemic inflammatory markers**	
The systemic cytokine profile was very similar to patient CRS, including increased levels of the murine CRP equivalent; G-CSF, GM-CSF, IFN-γ, IL-2, IL-3, and IL-6	The mice developed a broad cytokine response, including increased levels of a murine CRP analogue as well as IL1, IL-6, IL-10, and TNF-α.
**The role of monocytes**	
Host monocytes were a main source of released cytokines Monocytes express CD40 receptors, expression of CD40 ligand by CAR-T cells increased the severity of CRS CD40 ligation also increased the cytokine release by host monocytes. Expression of nitric oxide synthase was increased; aberrant nitric oxide production seemed to be directly involved in CRS pathophysiology probably due to its endothelial and/or vascular effects.	Monocytes were a major source of both IL-1 and IL-6. The monocytes expressed high levels of IL-1, IL-6, IL-8/CXCL8, CCL2, CCL8, and CXCL10. Dendritic cells also contributed to cytokine production.
**Effects of therapeutic interventions**	
IL-1RA protected against CRS mortality	IL-1RA/akinera protected from lethal neurotoxicity, this was a unique effect. Tocilizumab also reduced CRS mortality. CRS treatment did not influence the antileukemic effect of CAR-T cells.

**Table 4 jcm-10-05190-t004:** Major symptoms of CRS after organ involvement.

Organ System	Symptoms
**Constitutional**	Fever ± general malaise, fatigue
**Skin**	Rash
**Gastrointestinal**	Nausea, vomiting, diarrhea, anorexia
**Muscle, skeletal**	Myalgia, arthralgia
**Respiratory**	Tachypnea, hypoxemia
**Cardiovascular**	Tachycardia, dilated pulse pressure, hypotension, heart failure
**Coagulation**	Elevated D-dimer, hypofibrinogenemia, bleeding
**Renal**	Uremia
**Liver**	Transaminase increase, bilirubin increase
**Neurological**	Headache, change of mental status, confusion, delirium, aphasia, hallucinations, tremor, epileptic seizures

**Table 5 jcm-10-05190-t005:** Main differential diagnosis and clinical characteristics for CRS.

Differential Diagnosis	Clinical Characteristics
**Sepsis**	Sepsis can cause fever, hypotension, and respiratory complications. The evaluation for infection should include adequate microbiological diagnostics including blood cultures. It will often be necessary to initiate empirical antibiotic therapy.
**Disease progression**	Rapid progression of underlying malignancy can cause fever and a clinical, metabolic image similar to CRS.
**Tumor lysis syndrome**	The direct decay of malignant cells, especially in lymphoid malignancies, can cause metabolic disorders, with laboratory and clinical findings similar to CRS.
**Heart failure**	Cardiac failure due to cardiomyopathy, ischemic heart disease or pericardial effusion, may produce a clinical picture with respiratory failure as in severe CRS.
**Venous** **thromboembolism**	Clinical features of pulmonary embolism (PE) and deep vein thrombosis (DVT) such as dyspnea, hypoxia, hypotension, peripheral edema and swelling in the extremities may resemble CRS. Image diagnostics for this purpose may be relevant for diagnostic clarification.
**Acute respiratory distress syndrome (ARDS)**	Respiratory problem is the dominant symptom, with fluid accumulation in the lung tissue that can produce characteristic radiological changes.
**Allergic reaction/anaphylactic reaction**	Allergic reactions including severe drug reactions can cause fever, rash, capillary leakage and dyspnea. An overview of recent changes in the drug regimen should therefore be reviewed in case of suspected CRS.
**Hemophagocytic lymphohistiocytosis (HLH)**	HLH is a hyperinflammatory syndrome that shares common features and is likely related to CRS. Both by CRS and HLH are macrophage activation and cytokine storm.

**Table 6 jcm-10-05190-t006:** Classification of CRS severity according to the Lee criteria [73].

GRADE	SYMPTOMS
**GRADE 1**	Fever ≥ 38.0 °C. Symptoms are not life-threatening and require only symptomatic treatment (e.g., fever, nausea, headache, muscle pain, fatigue)
**GRADE 2**	Fever ≥ 38 °C. Symptoms need and respond to moderate measures: (i) oxygen requirements < 40% (≤6 L/minute), (ii) hypotension responding to IV fluid or low dose vasopressor1, or (iii) Grade 2 organ toxicity
**GRADE 3**	Fever ≥ 38 °C. Symptoms need and respond to aggressive measures: (i) oxygen demand ≥ 40% (≥6 L/minute), (ii) hypotension requiring high dose or multiple vasopressors ^1^, (iii) Grade 3 organ toxicity ^2^, or (iv) Grade 4 transaminitis ^3^
**GRADE 4**	Fever ≥38 °C. Life-threatening symptoms In need of CPAP, BiPAP or ventilator support, or Grade 4 organ toxicity ^2^
**GRADE 5**	Mors

^1^ High dose vasopressors: noradrenaline ≥ 20 mcg/min, dopamine ≥ 10 mcg/kg/min, adrenaline ≥ 10 mcg/min, phenylephrine ≥ 200 mcg/min. ^2^ Organ toxicity as defined by the *Common Terminology Criteria for Adverse Events* (CTCAE) v5.0. ^3^ Transaminitis as defined by CTCAE v5.0. Abbreviations: BiPAP, bilevel positive airway pressure; CPAP, continuous positive airway pressure.

**Table 7 jcm-10-05190-t007:** Classification of CRS severity according to the ASTCT criteria [74].

CRS Parameter	Grade 1	Grade 2	Grade 3	Grade 4
**Fever**	Fever ≥ 38 °C	Fever ≥ 38 °C	Fever ≥ 38 °C	Fever ≥ 38 °C
**WITH**				
**Hypotension**	None	Not requiring vasopressors	Requiring a vasopressor with or without vasopressin	Requiring multiple vasopressors (excluding vasopressin)
**AND/OR**				
**Hypoxia**	None	Requiring low-flow nasal cannula ^1^ or blow-by	Requiring high-flow nasal cannula ^1^, facemask, nonrebreather mask, or Venturi mask	Requiring positive pressure (e.g., CPAP, BiPAP, intubation or mechanical ventilation)

^1^ Low-flow nasal cannula is defined as oxygen delivered at ≤6 L/min. Low flow also includes blow-by oxygen delivery, sometimes used in pediatrics. High-flow nasal cannula is defined as oxygen delivered at >6 L/min. Abbreviations: BiPAP, bilevel positive airway pressure; CPAP, continuous positive airway pressure.

## Data Availability

No new data were created or analyzed in this study. Data sharing is not applicable to this article.

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
