# Peer review of "Cytokine Release Syndrome in the Immunotherapy of Hematological Malignancies: The Biology behind and Possible Clinical Consequences"

_jcm, 2021, doi:10.3390/jcm10215190_

Round 1

Reviewer 1 Report

The review is very interesting because actually, we are using immunotherapy in the treatment of many hemato oncological diseases. New drugs are introduced in chemotherapy protocols and it is important to know the pathogenesis and treatment of cytokine release. Moreover, in actual conditions determined by COVID 19 pandemic, this topic is important to be known because it is the main cause of complication in COVID-19. This article brings us all the information regarding this topic. So it is very interesting and could be one starting point in future research. It is well written and very well documented. 

Author Response

Although COVID-19 disease and cytokine storm are beyond the limitation for this review, we agree that this topic is interesting and at the present time highly relevant for the subject. Therefore, we have included a statement regarding COVID-19 in the introduction part of the paper.

Reviewer 2 Report

Dear Authors,

This is a very interested paper summarizing the background and management of cytokine release syndrome. The paper is well written, however I have some minor comments:

  1. Please review carefully the text for repetitions and rephrase senstences like: Macrophages care central in the development of CRS [30], and upon activation, mac-148 rophages release a wide range of cytokines including both interleukins, chemokines, and 149 immunoregulatory mediators (Figure 1). (line 148-149). This is just one example of many sentences with the repetitions of specific words within it.
  2. The data in Table 1 shows cytokine profile increased in advance CRS and  with no differences between low and advance stage of CRS. There are several cytokines listed in both groups, which you described as "conflicting results". Please elaborate more on this subject and discuss the research behind this conflicting results.

Author Response

We are thankful for the comments from the reviewer, which clearly has helped us improve the present manuscript. We have again read the manuscript an d tried to avoid unnecessary repetition and statements.

Furthermore, we have elaborate and clarified the presentation in Table 1, and special emphasize the cytokines, mainly IL-5 and IL-6, with conflict results in the study of CRS.